# The Use of ^18^F-FET-PET-MRI in Neuro-Oncology: The Best of Both Worlds—A Narrative Review

**DOI:** 10.3390/diagnostics12051202

**Published:** 2022-05-11

**Authors:** Tineke van de Weijer, Martijn P. G. Broen, Rik P. M. Moonen, Ann Hoeben, Monique Anten, Koos Hovinga, Inge Compter, Jochem A. J. van der Pol, Cristina Mitea, Toine M. Lodewick, Arnaud Jacquerie, Felix M. Mottaghy, Joachim E. Wildberger, Alida A. Postma

**Affiliations:** 1Department of Radiology and Nuclear Medicine, Maastricht University Medical Center+, P.O. Box 5800, 6202 AZ Maastricht, The Netherlands; rik.moonen@mumc.nl (R.P.M.M.); jochem.vander.pol@mumc.nl (J.A.J.v.d.P.); cristina.mitea@mumc.nl (C.M.); t.lodewick@mumc.nl (T.M.L.); arnaud.jacquerie@mumc.nl (A.J.); fmottaghy@ukaachen.de (F.M.M.); j.wildberger@mumc.nl (J.E.W.); l.jacobi@mumc.nl (A.A.P.); 2School of Nutrition and Translational Research in Metabolism (NUTRIM), P.O. Box 616, 6200 MD Maastricht, The Netherlands; 3Department of Neurology, Maastricht University Medical Center+, P.O. Box 5800, 6202 AZ Maastricht, The Netherlands; martijn.broen@mumc.nl (M.P.G.B.); m.anten@mumc.nl (M.A.); 4School for Oncology and Reproduction (GROW), P.O. Box 616, 6200 MD Maastricht, The Netherlands; ann.hoeben@mumc.nl (A.H.); koos.hovinga@mumc.nl (K.H.); inge.compter@maastro.nl (I.C.); 5Department of Medical Oncology, Maastricht University Medical Center+, P.O. Box 5800, 6202 AZ Maastricht, The Netherlands; 6Department of Neurosurgery, Maastricht University Medical Center+, P.O. Box 5800, 6202 AZ Maastricht, The Netherlands; 7Department of Radiation Oncology (Maastro), Maastricht University Medical Center+, P.O. Box 5800, 6202 AZ Maastricht, The Netherlands; 8Cardiovascular Research Institute Maastricht (CARIM), P.O. Box 616, 6200 MD Maastricht, The Netherlands; 9School for Mental Health and Neuroscience (MHeNs), Maastricht University Medical Center+, P.O. Box 616, 6200 MD Maastricht, The Netherlands

**Keywords:** glioma, FET-PET, MRI, PET-MRI

## Abstract

Gliomas are the most frequent primary tumors of the brain. They can be divided into grade II-IV astrocytomas and grade II-III oligodendrogliomas, based on their histomolecular profile. The prognosis and treatment is highly dependent on grade and well-identified prognostic and/or predictive molecular markers. Multi-parametric MRI, including diffusion weighted imaging, perfusion, and MR spectroscopy, showed increasing value in the non-invasive characterization of specific molecular subsets of gliomas. Radiolabeled amino-acid analogues, such as 18F-FET, have also been proven valuable in glioma imaging. These tracers not only contribute in the diagnostic process by detecting areas of dedifferentiation in diffuse gliomas, but this technique is also valuable in the follow-up of gliomas, as it can differentiate pseudo-progression from real tumor progression. Since multi-parametric MRI and 18F-FET PET are complementary imaging techniques, there may be a synergistic role for PET-MRI imaging in the neuro-oncological imaging of primary brain tumors. This could be of value for both primary staging, as well as during treatment and follow-up.

## 1. Introduction

Gliomas are the most prevalent primary brain tumors [1]. Most gliomas are sporadic, with only few cases showing a relationship to genetic syndromes, including neurofibromatosis type 1. Gliomas are subdivided into low grade gliomas (LGG) and high grade gliomas (HGG) [2]. The histomolecular subtypes are associated with the clinical behavior of these tumors, and hence, the clinical outcome. Current treatment schedules are guided by these histomolecular subtypes as reflected in the new WHO classification, which was revised in 2021 [3]. Here, diffuse gliomas (LGG) are subdivided partly based upon molecular markers into: astrocytomas, IDH-mutated tumors without 1p/19q co-deletion, oligodendrogliomas, IDH-mutated tumors with a 1p/19q-codeletion, and IDH-wildtype astrocytomas (WHO 2021 5th edition). The behavior of LGG is variable, with stable disease in some of these gliomas for a prolonged period, whilst other LGGs may progress within a short period of time to a HGG. Therefore, the median survival rate has a broad range of 1–15 years. 

The treatment of gliomas is based on the management of the symptoms and the reduction, or if possible, the resection of the tumor. The optimal treatment is controversial, and ranges from long-term follow-up to surgery, depending on the size and location of the tumor and the tumor grade. In some cases, radiological diagnosis, especially in low grade tumors, can limit debility from invasive tissue analysis. However, in most cases, this is not sufficient, and the tissue biopsy or surgical resection is needed for an accurate staging and subtyping of the gliomas. Depending on the subtype, grading, and Pignatti criteria, an additional treatment with sequential radiotherapy and chemotherapy is indicated [4]. Overall, surgery is associated with an increased survival rate. In some cases, eloquent location of the tumor prohibits total surgical debulking, and only a partial debulking or histological biopsy can be performed. Since intra-tumoral heterogeneity is expected, histopathological analysis is important to collect tissue in the regions with the worst histopathological profile for an accurate classification of these gliomas and optimal treatment, accordingly. 

Neuro-imaging plays an important role in the characterization of gliomas, biopsy planning, and surgical resection [5]. Magnetic resonance imaging (MRI) is the standard for the diagnostic imaging of gliomas [6]. However, positron emission tomography (PET) imaging with amino-acid tracers is being used more frequently in neuro-oncology as an add-on for MRI [7]. 

As both techniques are complementary and essential for the follow-up of the patient, there may be a possible role for hybrid imaging with PET-MRI within neuro-oncology for the staging and follow-up of disease. Both modalities and their integration within hybrid imaging with PET-MRI are discussed in this narrative review.

## 2. MRI

MRI is the standard for the imaging of gliomas. Here, at least a T2, FLAIR, and a 3D T1-weighted series, before and after administration of a gadolinium-based contrast agent, should be acquired. These sequences still have limitations with respect to the delineation of the tumor, the tumor grading, and the differentiation between (radiation) necrosis versus progression after radiotherapy [8,9]. For instance, no contrast enhancement is found in one-third of the grade II and III tumors [10] (see Figure 1). This fact carries a higher risk of miss-sampling at biopsy, resulting in an undergrading of up to 28% of WHO grade III tumors [11]. The addition of multi-parametric MRI with diffusion weighted imaging (DWI), perfusion weighted imaging (PWI), and/or magnetic resonance spectroscopy (MRS) has improved the grading or these tumors considerably, with the opportunity to non-invasively obtain information on the histomolecular profile. 

With diffusion weighted imaging (DWI), the movement of water molecules can be studied. This sequence provides information on the cellularity of the tumor, expressed as the so-called apparent diffusion coefficient (ADC). A lower ADC is associated with a higher tumor grade [12]; the specificity however, remains relatively low (sensitivity 97.6%, specificity 53.1% [13]). Furthermore, ADC can have value in the prediction of the molecular profile [14,15,16]. ADC showed a sensitivity of 83–87% and a specificity of 55–60% for the prediction of the IDH1/2 mutation status [16,17].

Perfusion imaging provides information on tissue perfusion and permeability, which corresponds to the microvascular proliferation and leakage [17,18,19,20]. One of the most frequently used parameters is the relative “cerebral blood volume” (rCBV), a measurement wherein the tumor is compared to the contralateral white matter (or representative white matter) (see Figure 2). The rCBV in HGGs is elevated, and is 1.9 on average, but only 1.3 on average in LGGs [21]. However, again, the variation between the different tumors is large, and tumors tend to be very heterogeneous, creating a large overlap in the rCBV values between LGG and HGG. Therefore, the differentiation between LGG and HGG is difficult based on PWI alone. 

PWI can also be used for the prediction of the IDH1/2-mutation, with a sensitivity and specificity of 0.77 and 0.88 [22,23]. A rCBVmax > 2.35 is associated with the IDH-wildtype status and hence, a poor prognosis for the patient [14]. 

PWI is currently mainly used in the follow-up of patients after therapy, where the rCBV proves to be the most validated parameter [16,24,25]. For the differentiation of true progression versus pseudo-progression, PWI can be of value. Pseudo-progression is a self-resorbing focal contrast enhancement on MRI [26]. This has an incidence of about 36% in patients with glioblastoma (GBM) after treatment with radiotherapy in combination with temozolomide [26]. It is caused by an increase in vascular permeability and inflammatory changes [27]. This is a challenging paradigm, and often clinicians have to resort to tissue diagnosis for the differentiation of pseudo-progression from real progression to prevent the premature cessation of efficacious therapeutic agents. Hence, there is a need for more radiological information to limit the need of biopsy for diagnosis. PWI might be valuable for improving the differentiation of pseudo-progression from real progression, when compared to conventional MRI. A recent meta-analysis showed a pooled sensitivity and specificity of 92% and 85%, respectively, for PWI using dynamic contrast-enhanced (DCE) perfusion imaging. This is promising; however, there is a need for standardization of the threshold for the rCBV and a uniformity in data acquisition [28]. 

“Pseudo-response”, on the other hand, is a rapid regression of focal enhancement without a true remission of the tumor. It is mostly caused by the anti-angiogenetic effects of bevacizumab (BEV). In some cases, PWI can show areas with persisting increased perfusion, despite the loss of enhancement after contrast administration. This could indicate pseudo-response. However, the absolute values are not consistent, and data acquired are variable; hence, the identification of pseudo-response with PWI remains difficult [29,30].

The value of DWI in pseudo-response and pseudo-progression is limited. Although the ADC values are associated with progressive disease, the use of the ADC is controversial for the differentiation of pseudo-progression versus real progression in glial tumors after therapy. This is caused by a large intrinsic heterogeneity, with vast differences within the regions with high cellularity versus areas with necrosis, edema, and micro-bleeds after therapy. This makes the differentiation based solely on ADC impossible [31].

Lastly, MR spectroscopy (MRS) can aid in the diagnosis of glial tumors. With MR spectroscopy, specific molecules can be identified and quantified [32,33,34,35,36]. A specific and promising MRS measurement is the measurement of 2-hydroxyglutarate (2HG). The accumulation of 2HG in glial tumors is a consequence of the IDH1/2-mutation [37]. In a recent meta-analysis of Suh et al [38], 14 studies were compared. Consistently, a higher 2HG peak was found in the patients with the IDH1/2 mutation [38]. It was also suggested that the measurement of 2HG with MRS could aid in the evaluation of targeted treatment with IDH-inhibition with, for instance, ivosidenib and vorasidenib [39].

The measurement of 2HG is strongly dependent on the cellularity of the tumor [40] and the tumor volume [41]. This may limit the implementation of this method in current daily practice. 

Thus, although the results of the studies using DWI, PWI, and MRS are promising, an unambiguous standardized clinical protocol is still missing for the current daily practice of the implementation of these techniques. The threshold values used in the literature are variable, and the experience with the MRS techniques is limited [42]. 

In current clinical practice, based on this evidence, besides the standard (T2, FLAIR and a 3D T1-weighted series with and without contrast), a DWI and PWI could be considered to be added to the protocol. PWI will mainly contribute to the differentiation of pseudo-progression in FUP of gliomas after radiotherapy. Here, a rCBVmax > 2.35 may be used as a cut-off. MRS is technically challenging, and therefore, is not recommended as a clinical standard. MRS may be valuable in selective cases where histological biopsy is not possible and IDH-mutation status needs to be confirmed.

## 3. 18F-Fluoroethyl-L-Tyrosine, 18F-FET-PET

For the imaging of gliomas, radiolabeled amino-acid analogues have been used. In this context, ^18^F-FET, 3,4-dihydroxy-6-^18^F-Fluoro-l-phenylalanine (^18^F-DOPA) and ^11^C-Methionine are the most well-known tracers. However, the most widely clinical used and available tracer is O-(2-^18^F-fluoroethyl)-l-tyrosine (^18^F-FET). Therefore, we will mainly focus on ^18^F-FET. The accumulation of ^18^F-FET in gliomas is due to the overexpression of the LAT2-L-transporter. This overexpression leads to an increased tracer uptake in tumor tissue compared to the background [43]. Here, a mean and maximal standardized uptake value in the tumor is measured and compared with the contralateral “normal” brain tissue. From these data, the tumor-to-brain ratio (TBR_mean_, TBR_max_) is calculated [44]. The use of ^18^F-FET PET in gliomas is recommended both by the European Association of Nuclear Medicine (EANM), as well as the European Association of Neuro-Oncology (EANO) for the detection of viable tumor tissue after therapy, as well as for the differentiation of gliomas versus non-glial brain tumors [44,45]. 

For the differentiation of HGG and LGG, *static* ^18^F-FET PET can be used, although an overlap is seen for the different WHO grades in static PET-imaging [46]. *Static* ^18^F-FET PET has a sensitivity of 71–80% and a specificity of 56–85% for the differentiation between HGG and LGG [44,47]. Employing *dynamic* ^18^F-FET can increase the accuracy [44,48]. The use of 4D imaging 50 min after injection allows a voxel-wise generation of a time-activity-curve (TAC) [43,45]. These TACs show a clear difference between LGG and HGG. More specifically, HGG often show a descending curve, while LGG remains at a plateau, or even increases in activity over time (see Figure 1) [44,47]. Furthermore, there are indications that the *IDH1/2* mutation status can be determined non-invasively through ^18^F-FET PET [49,50]. 

Although ^18^F-FET PET is accurate for the different subtypes of gliomas, the accuracy is low for oligodendrogliomas, due to high tracer accumulation both in grade II and grade III oligodendrogliomas [51,52]. Other possible pit-falls of ^18^F-FET PET comprise benign pathologies, including infarction, inflammation, or insults [53,54,55]. These pathologies can increase local tracer accumulation on the ^18^F-FET PET (see Figure 3D–H). Therefore, for the interpretation of the ^18^F-FET PET and the analysis of the dynamic images, a combination with MRI is essential to achieve a good differential diagnosis. 

It is clear that ^18^F-FET PET plays an important role in the differentiation between progression and pseudo-progression, as a consequence of (chemo-)radiotherapy (see Figure 2 and Figure 4). Galldiks et al. showed that the tracer accumulation of ^18^F-FET was significantly lower in areas of pseudo-progression, compared to areas with real progression, with a performance of ^18^F-FET-PET vastly superior to DCE-MRI and PWI [56]. Other studies confirmed these findings [57,58,59]. The ^18^F-FET PET also seems to produce valuable information in the non-invasive determination of the molecular subtype, prognosis, and treatment (mainly for the determination of the biological tumor volume) and for the interpretation of early changes after treatment, when compared with MRI [60]. Since patients that do not benefit from treatment can possibly be identified in an early stage with ^18^F-FET PET, treatment regimens can be changed more rapidly, reducing the treatment burden and optimizing individualized treatment regimens.

For the differentiation of pseudo-response, ^18^F-FET PET can be of value, as the tracer uptake is not influenced by neo-vascularization. Using ^18^F-FET PET rather than MRI as a tool for the evaluation of the treatment of anti-angiogenesis inhibitors, such as BEV, can reduce overtreatment and its associated side-effects (for example see Figure 5) [56,61]. In patients with a HGG, ^18^F-FET PET could possibly even predict the outcome of treatment with BEV [62,63]. 

## 4. ^18^F-FET-PET-MRI

The pathologic findings on multi-parametric MRI, including rCBV, do not correspond anatomically with the areas of increased tracer accumulation at ^18^F-FET-PET [64]. This is due to the fact that both modalities depict different biological processes [65]. The combination of multi-parametric MRI with ^18^F-FET-PET could improve both the non-invasive characterization of these diffuse infiltrating gliomas, as well as the guidance of stereotactic biopsies. There are several studies that prove the value of both multi-parametric, MRI as well as ^18^F-FET-PET, in the prediction of the behavior of these glial tumors [66]. It is possible that combined ^18^F-FET-PET-MRI is better than ^18^F-FET-PET alone; however, this has not yet been systematically studied. 

The hybrid PET-MRI is a modality in which the PET detector is directly integrated into the MRI scanner, making simultaneous PET and MRI acquisition possible. Integrated PET-MRI has the advantage that the PET and MRI images are directly and accurately fused. This is of great importance in neuro-navigation. Song et al. showed that simultaneous ^18^F-FET-PET and MRI improve tumor delineation and spatial correlation in gliomas [67]. All integrated PET-MRI systems are currently equipped with a 3 Tesla MRI and a digital PET-detector system. However, the images of these systems cannot be compared one on one with separate state-of-the-art MRI and PET scanners, due to technical compromises. Nonetheless, the images are still of high diagnostic quality. 

Taking in account patient comfort, one could consider the preferential use of PET-MRI in this fragile patient population. By making one integrated scan with the PET-MRI compared to two separate scans, the patient burden can be reduced [68]. The standard ^18^F-FET-PET protocol is of sufficient length to simultaneously acquire a multi-parametric MRI of the brain. The use of gadolinium contrast does not interfere with the PET acquisition. The hybrid PET/MRI, therefore, almost halves the total acquisition time. Hybrid PET-MRI acquisition as a one-stop shop spares the patient from an additional hospital visit. This, in our experience, vastly increases patient comfort and satisfaction.

The addition of a ^18^F-FET PET scan to the standard MRI is at the first view expensive, with a range of EUR 1650 to EUR 3000, on top of the costs of an MRI scan. However, a good treatment plan based on adequate imaging with both PET and MRI has proven to be cost-effective in multiple studies [43,69]. Additionally, for response evaluation, ^18^F-FET PET reduces costs by avoiding unnecessary treatment and follow-up [43]; the implementation of ^18^F-FET PET is also is cost-effective in the primary diagnosis of the tumor. The study of Heinzel et al. showed, for instance, that the combined use of PET and MRI resulted in an increase of 18.5% in the likelihood of a correct diagnosis. The incremental cost-effectiveness ratio for one additional correct diagnosis using FET PET was EUR 6405 for the baseline scenario and EUR 9114 for the scenario based on higher disease severity [43,69]. This strongly outweighs the costs of the implementation of the ^18^F-FET PET in clinical practice. Hence, there seems to be a future for the further development and implementation of ^18^F-FET-PET-MRI imaging in neuro-oncological imaging, where both modalities are complementary to each other.

## 5. Conclusions

There are many developments in the field of multi-parametric MRI (ADC, rCBV, and MRS) that may contribute to a better characterization and follow-up of gliomas. However, the further development of good clinical standards is needed to further implement these techniques clinically. For the differentiation of pseudo-progression and true progression, ^18^F-FET-PET has been proven to be of great value. The use of ^18^F-FET-PET and multiparametric MRI are complementary, and the combined use at primary staging could lead to a better guidance of stereotactic biopsies/surgeries, a better prediction of prognosis, and a better prediction of therapy response. Furthermore, the combined use will improve the detection of tumor progression during follow-up. Here, hybrid PET-MRI imaging may be of value as the anatomic correlation of PET and MRI is improved, which is of vital importance for the guiding of biopsies and surgeries. Furthermore, it will improve patient comfort by reducing hospital visits and scan time. Therefore, it may be expected that hybrid ^18^F-FET-PET-MRI imaging may play a larger role in the future of neuro-oncologic imaging.

## Figures and Tables

**Figure 1 diagnostics-12-01202-f001:**
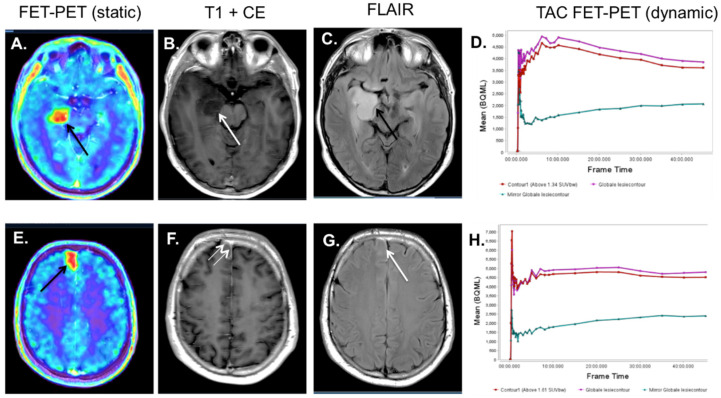
Patient presenting with an epileptic seizure. Initially, based on the MRI, the patient was suspected to have a LGG, as there was a lack of contrast enhancement. However, in the work-up, a ^18^F-FET-PET-MRI was acquired. The PET-MRI shows a glial tumor in the right mesial temporal lobe on the FLAIR (black arrow, (**C**)), without contrast enhancement, with an intense tracer accumulation centrally in the lesion (black arrow, (**A**)). (**B**) shows the MRI T1 after contrast administration, showing no enhencement at the site of the leasion (white arrow). On the TAC of the dynamic ^18^F-FET-PET (**D**), a descending curve is visible in red/purple, making the lesion more suspicious of an HGG. At biopsy, an astrocytoma WHO grade III was found, IDH-wt, without MGMT promotor methylation or 1p/19q co-deletion. On the 18F-FET-PET-MRI, a second lesion was identified in the right frontal lobe, with increased tracer uptake (black arrow, (**E**)). Initially, this was missed on the MRI. Here, subtle contrast enhancement of the meninges is visible (white arrows, (**F**)), accompanied by a slightly increased signal intensity on the FLAIR (white arrow, (**G**)). The lesion shows a plateau on the TAC (red/purple), with a gradual decrease at the end of the curve (**H**). At biopsy, a astrocytoma WHO grade III was found.

**Figure 2 diagnostics-12-01202-f002:**
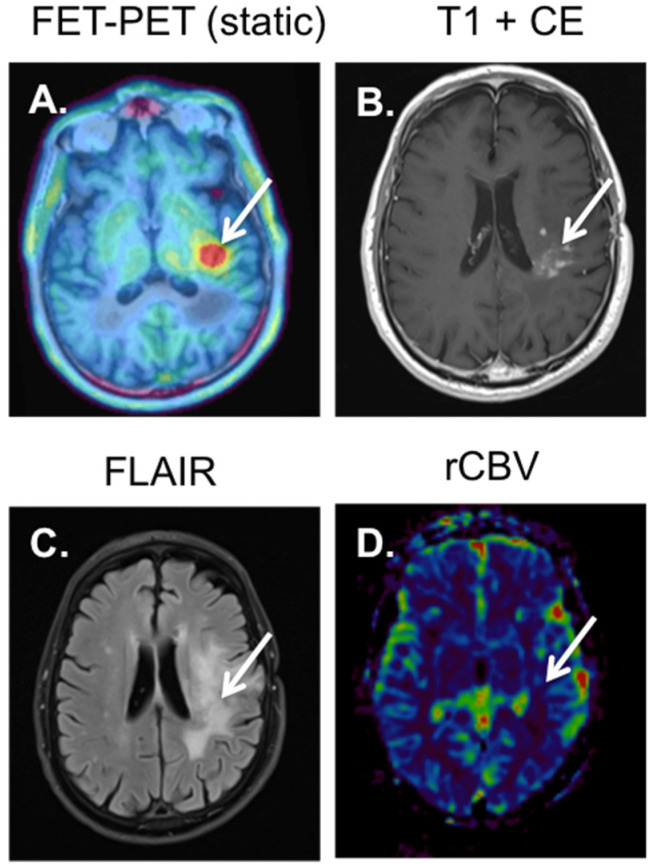
Patient with an astrocytoma grade III, IDHwt that was treated with radiotherapy and temozolomide. Five months after treatment, persisting contrast enhancement is visible in the left parietal lobe (white arrow (**B**)), with accompanying gliotic changes on FLAIR (white arrow (**C**)), without an increase in perfusion. See relative rCBV map (white arrow (**D**)). The ^18^F-FET-PET (**A**) shows an increased tracer accumulation in the area of contrast enhancement with a TBR_max_ of 2.5 (>2.0), indicating progressive disease (see white arrow (**A**)).

**Figure 3 diagnostics-12-01202-f003:**
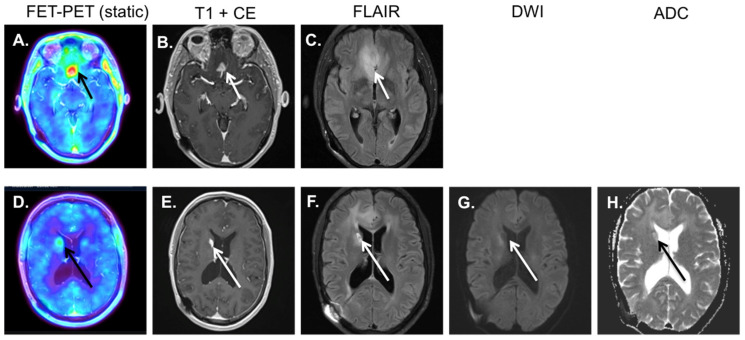
Patient with an anaplastic astrocytoma, 3 months after radiotherapy and temozolomide treatment. On the MRI, a persistent contrast enhancement is visible in the lesion (frontal lesion marked with white arrow (**B**)), with increased signal intensity on the FLAIR (see white arrow (**C**)). Based on MRI alone, pseudo-progression cannot be distinguished from real progression. The ^18^F-FET-PET shows a strongly increased accumulation of the tracer, typical for viable tumor tissue (black arrow ^18^F-FET-PET image (**A**)). This is confirmed by biopsy. Furthermore, the ^18^F-FET-PET shows a second area with increased tracer accumulation at the caput caudate nucleus on the right side (black arrow ^18^F-FET-PET image (**D**)). However, this was a false positive, as the MRI (white arrows (**E**,**F**)) and DWI/ADC (white and black arrow on (**G**,**H**)) show clear signs of ischemia, possibly secondary to radiotherapy. This example shows how these modalities can be complementary to each other.

**Figure 4 diagnostics-12-01202-f004:**
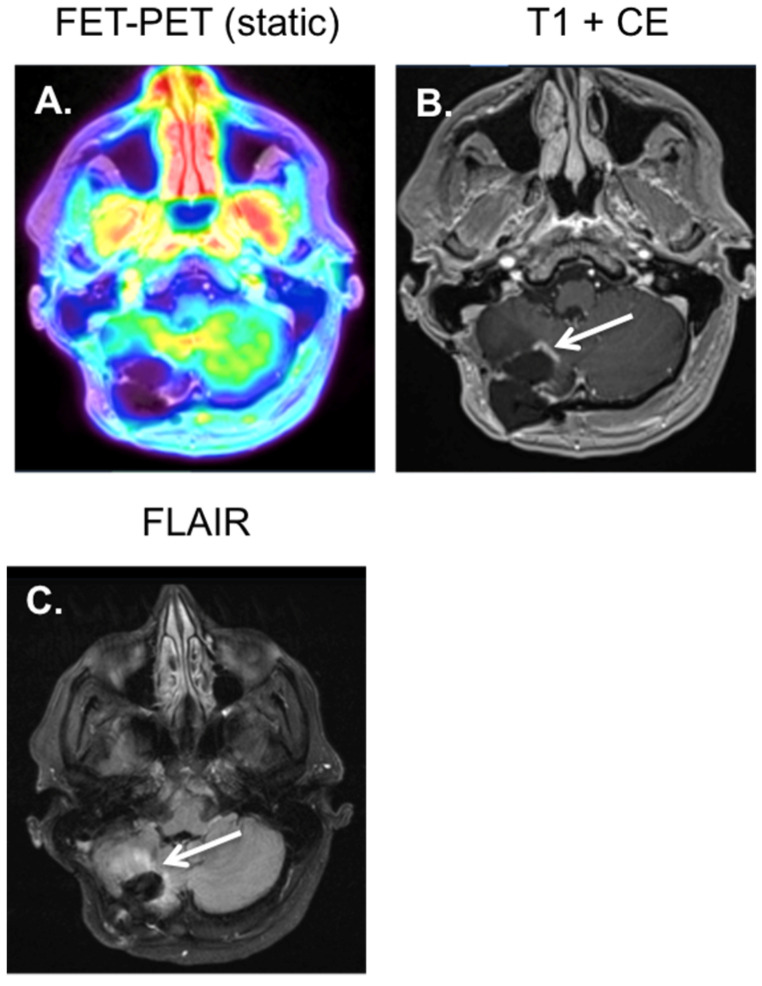
Patient with a GBM in the right cerebellar hemisphere, after resection and concurrent radiotherapy and temozolomide treatment. Three months after treatment, an irregular nodular contrast enhancement is visible on the MRI near the edges of the resection site (see white arrow image (**B**)), with gliotic changes around this area on the FLAIR (see white arrow image (**C**)). The ^18^F-FET-PET (**A**) only showed a low tracer uptake in the area of contrast enhancement (TBR_max_ = 1.4 < 2.0), which was interpreted as radio-necrosis.

**Figure 5 diagnostics-12-01202-f005:**
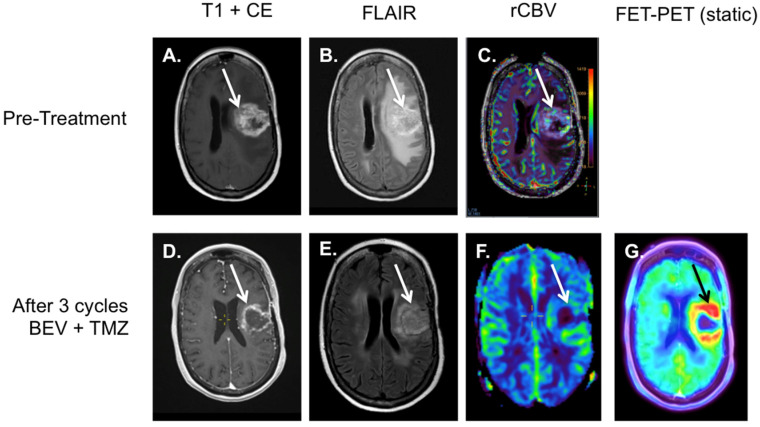
Patient with a GBM in the left frontal lobe that was treated with debulking and concurrent chemoradiation. Patient presented with progressive neurological decay. On the first post-treatment MRI, a persistent mass with contrast enhancement was found (white arrow image (**A**)), with edema (white arrow image (**B**)) and mass effect causing midline shift (**B**) and areas with increased perfusion on the rCBV at the resection margins (white arrow image (**C**)). Patient was subsequently treated with 3 cycles of BEV and temozolomide, with a clear decrease in contrast enhancement on MRI (white arrow, image (**D**)), and a decrease in mass effects, edema, and gliosis (white arrow image (**E**)). Perfusion did not show any areas with persistent increased perfusion on the rCBV map (white arrow image (**F**)). However, the ^18^F-FET-PET did show a diffuse elevated tracer accumulation at the height of the resection margins (black arrow image (**G**)), illustrative of pseudo-response.

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
