# Peer review of "The Use of 18F-FET-PET-MRI in Neuro-Oncology: The Best of Both Worlds—A Narrative Review"

_diagnostics, 2022, doi:10.3390/diagnostics12051202_

Round 1

Reviewer 1 Report

I wish to congratulate the authors since they identified a novel and important topics in nuclear medicine/neurooncology. I do recognize that this manuscript has to be considered a narrative review, more than a systematic one: this should be underlined in the introduction and title.

All the reported clinical cases are very explicative and they will help the reader, just to evaluate the opportunities that this mixed imaging technique (PET-MRI) will ensure in the next future.

In the attached file, I have added some minor comments; all of them are related to the yellow-labelled sections of the manuscript

Author Response

Response to the reviewer comments

We would like to thank the reviewers for the appreciation of our work and the valuable and constructive comments to the manuscript. We believe that the manuscript has improved due to the suggested changes and we have addressed all comments in a point-to-point response below.

Reviewer #1:

I wish to congratulate the authors since they identified a novel and important topics in nuclear medicine/neuro-oncology. I do recognize that this manuscript has to be considered a narrative review, more than a systematic one: this should be underlined in the introduction and title.

Indeed it was written as a narrative review, and we have added this to the titel and the introduction.

All the reported clinical cases are very explicative and they will help the reader, just to evaluate the opportunities that this mixed imaging technique (PET-MRI) will ensure in the next future.

In the attached file, I have added some minor comments; all of them are related to the yellow-labeled sections of the manuscript

We have revised the manuscript according to the marked sections in the document. We have marked these sections in yellow. We think the manuscript has much improved, thank you!

Reviewer #2:

Thank you for the opportunity to review this manuscript. This is a well written manuscript with circumspect review of existing literature on MRI and PET imaging and combination thereof of both modalities in neuro-oncology. I do have few suggestions for the authors in regard to the manuscript.

  • I would recommend adding a few lines and citation in the introduction section about the difference in management and prognosis of primary CNS malignancies based of grading ranging from monitoring with serial imaging to debulking with chemo-radiation. Although, molecular markers and immunohistochemical testing of tissue sample are deemed gold standard for diagnosis, this is not always the case and radiological diagnosis especially in low grade tumors can limit debility from invasive tissue analysis.

Indeed in low grade glioma’s, monitoring can be an option, instead of active therapy. We have added a few sentences in the introduction. We have marked this section with green on page 2.

  • The authors have touched upon tumor progression and pseudo-progression in the manuscript. This is a challenging paradigm and often clinicians have to resort to tissue diagnosis. I would recommend that authors elaborate on the discussion between the difference between the two and emphasize on the utility of more radiological information to limit the need of biopsy for diagnosis, thereby limiting the morbidity in these patients.

We have tried to emphasize this in the section, pointing out the consequences of this challenge and urging the need for better radiological imaging. This section was marked in green on page 4.

  • I would also recommend adding arrows and marker to the imaging section for a easy read for readers

Thank you for the suggestion. We have changed the figures accordingly.

  • and lastly adding a conclusion section summarizing authors conclusion succinctly.

We have added a summarizing conclusion in the end, marked in green on page 9.

Reviewer 2 Report

Thank you for the opportunity to review this manuscript. This is a well written manuscript with circumspect review of existing literature on MRI and PET imaging and combination thereof of both modalities in neuro-oncology. I do have few suggestions for the authors in regard to the manuscript. I would recommend adding a few lines and citation in the introduction section about the difference in management and prognosis of primary CNS malignancies based of grading ranging from monitoring with serial imaging to debulking with chemo-radiation. Although, molecular markers and immunohistochemical testing of tissue sample are deemed gold standard for diagnosis, this is not always the case and radiological diagnosis especially in low grade tumors can limit debility from invasive tissue analysis. The authors have touched upon tumor progression and pseudo progression in the manuscript. This is a challenging paradigm and often clinicians have to resort to tissue diagnosis. I would recommend that authors elaborate on the discussion between the difference between the two and emphasize on the utility of more radilogical information to limit the need of biopsy for diagnosis, thereby limiting the morbidity in these patients. I would also recommend adding arrows and marker to the imaging section for a easy read for readers and lastly adding a conclusion section summarizing authors conclusion succinctly.

Author Response

(The authors gave the same response as above.)
